# A big data analysis algorithm for massive sensor medical images

Sarah A. Alzakari[1], Nuha Alruwais[2], Shaymaa Sorour[3], Shouki A. Ebad[4], Asma Abbas Hassan Elnour[5] and Ahmed Sayed[6]

[1] Department of Computer Sciences, Princess Nourah bint Abdulrahman University, Riyadh, Saudi Arabia
[2] Department of Computer Science and Engineering, King Saud University, Riyadh, Saudi Arabia
[3] Department of Management Information Systems, King Faisal University, Al-Ahsa, Saudi Arabia
[4] Department of Computer Science, Faculty of Science, Northern Border University, Arar, Saudi Arabia
[5] Computer Science Department, King Khalid University, Riyadh, Saudi Arabia
[6] Research Center, Future University in Egypt, New Cairo, Egypt



Corresponding author
Shouki A. Ebad,
shouki.abbad@nbu.edu.sa

## ABSTRACT

Big data analytics for clinical decision-making has been proposed for various clinical sectors because clinical decisions are more evidence-based and promising. Healthcare data is so vast and readily available that big data analytics has completely transformed this sector and opened up many new prospects. The smart sensor-based big data analysis recommendation system has significant privacy and security concerns when using sensor medical images for suggestions and monitoring. The danger of security breaches and unauthorized access, which might lead to identity theft and privacy violations, increases when sending and storing sensitive medical data on the cloud. Our effort will improve patient care and well-being by creating an anomaly detection system based on machine learning specifically for medical images and providing timely treatments and notifications. Current anomaly detection methods in healthcare systems, such as artificial intelligence and big data analytics-intracerebral hemorrhage (AIBDA-ICH) and parallel conformer neural network (PCNN), face several challenges, including high resource consumption, inefficient feature selection, and an inability to handle temporal data effectively for real-time monitoring. Techniques like support vector machines (SVM) and the hidden Markov model (HMM) struggle with computational overhead and scalability in large datasets, limiting their performance in critical healthcare applications. Additionally, existing methods often fail to provide accurate anomaly detection with low latency, making them unsuitable for time-sensitive environments. We infer the extraction, feature selection, attack detection, and data collection and processing procedures to anticipate anomaly inpatient data. We transfer the data, take care of missing values, and sanitize it using the pre-processing mechanism. We employed the recursive feature elimination (RFE) and dynamic principal component analysis (DPCA) algorithms for feature selection and extraction. In addition, we applied the Auto-encoded genetic recurrent neural network (AGRNN) approach to identify abnormalities. Data arrival rate, resource consumption, propagation delay, transaction epoch, true positive rate, false alarm rate, and root mean square error (RMSE) are some metrics used to evaluate the proposed task.

## INTRODUCTION

Analyzing sensor networks and medical images proves tedious because of the large amount of information. Current approaches are frequently ineffective in transmitting and making known decisions as viewed, especially in the current real-time patient health status management, which is fraught with challenges and more variables. Big data analytics seems able to accelerate the development of biological rationale to support evidence based on the therapy course of action across therapeutic areas. Nonetheless, complex parts still exist regarding data management, especially when transforming data sets polychained into a chronological series. This sometimes necessitates statistical methods for analyzing incomplete datasets, preparing datasets collected at different time points, and deciding the data treatment for the time series to be uniform. Also, the combination of various algorithms that were used in the earlier investigations led to further wastage of resources since a lot of calculations had to be done, and alteration of data was done all in a bid to reduce processing time. Moreover, if the window size is unwise for pattern detection in recognizing vital structures, detecting patterns may be devoid of extreme changes in the information, but too small patterns inside the data may be left behind. Security issues also come with unauthorized access, data, communication, execution manipulation, and interception of data, posing significant threats. It is, therefore, a critical but challenging task to uphold good security in sensor-based medical data. Such analysis of big data is necessary as a huge amount of sensor data about organ imaging and physiological systems should be utilized for the detection of medical conditions and real-time monitoring of the patient's health. By maximizing big data techniques, this research tackles the problem of dealing with unstructured medical information, gets relevant and salient features, and enhances the relatively impoverished work of anomaly detection in medical data.

Imaging studies such as computer tomography (CT), magnetic resonance imaging (MRI), and ultrasound imaging assist in diagnosing diseases present in the human body and are thus essential assets in medicine today. Medical imaging is used by practitioners and scientists to examine tissues and organs in the human body, collect relevant information concerning these organs and tissues through the use of such imaging, and devise treatment based on their clinical training (*Mansour et al., 2023*). There is also a substantial body of this kind of medical data, particularly images, which justifies the application of more recent approaches based on big data, AI, and ML.

The five primary attributes commonly referred to as the "5V" are volume (the total quantity of data created), variety (data from various categories), velocity (the rate at which data is generated), variability (the inconsistent nature of data), and veracity (the caliber of the data that is gathered) (*Tchito Tchapga et al., 2021*). On the other hand, big data (BD), or unstructured data, does not conform to the standard formats used for data processing. Large data collections that are too big to handle, store, or analyze using conventional

techniques are called big data. Without analysis, it is kept in storage. To turn this data into value, certain technologies and methodologies are needed because it is impossible to search through and analyze without a clear schema (*Batko & Ślkezak, 2022*). This is accomplished by various computational approaches, which increasingly use machine learning techniques. This shift is typified by the change in the data function from "passive" to "active." When data plays an "active" role, it supports or contradicts existing theories and generates new ones that may be investigated on a never-before-seen scale (*Rodrigues et al., 2021*). Complex patterns in imaging data may be automatically recognized by AI systems with good performance. Various techniques have led to the discovery of several applications that quickly advance the discipline. Magnetic resonance imaging (MRI) is the most widely used technique for identifying brain tumors. Medical imaging uses MRIs to show aberrant bodily tissues (*Arabahmadi, Farahbakhsh & Rezazadeh, 2022*). The Internet of Things (IoT)-based system plays a critical role in the widespread use of telemedicine in recent times, including remote detection, prediction, monitoring, recovery, and therapy. Recently, IoT has been applied in the medical field with interest in creating smart technologies, such as medical tracking systems, medical diagnosis, prediction, and treatment systems, and smart healthcare (*Awotunde, Ogundokun & Misra, 2021*; *Sekuboyina et al., 2021*).

Numerous healthcare applications are possible with blockchain technology, such as cloud storage of insurance data, clinical trial data, electronic medical records for sharing and storing, monitoring devices, and mobile health apps. Blockchain is stable because anybody seeking to edit a block after it has been added to the chain would need to refigure the changed block and all subsequent blocks, which would require an unimaginably high computer power (*Chattu, 2021*). Because of this, IoT systems provide a greater security risk than earlier computer paradigms, and traditional methods of resolving security issues may not be effective in these cases. Because of this, maintaining and managing the IoT system's security is a highly challenging task. Deep learning offers predictable temporal and spatial inference efficiency while operating in resource-related and unpredictable situations. It is also essential for maintaining the amount of resources, which might be used to lessen the chores associated with inference at shortened computation times (*Abbas et al., 2022*; *Asha et al., 2023*; *Kim, Shah & Kim, 2022*). Combining two previously studied topics, scheduling, and power control, joint scheduling and power control approaches have been devised to successfully perform multi-objective optimization (*Huang et al., 2023*).

Designers assess the efficacy of our suggested methodology through simulation. Some advantages of the monitoring system are the ability to track well-being, the home treatment function, and the early identification of health conditions. The relatively new network idea of software-defined networking has gained prominence (*Singh & Chatterjee, 2023*; *Mohanty, Sahoo & Kabat, 2023*; *Abououf et al., 2023*; *Uppal et al., 2022*). To develop a system that can quickly analyze large amounts of data and accurately identify abnormalities in real time, this project investigates the combination of state-of-the-art machine learning algorithms with sensor-generated data. Our goal is to develop a system that, *via* process optimization at every level, not only lowers false positives, a crucial element in healthcare applications, but also enhances the accuracy of big data analysis in

medical picture anomaly detection. The effective implementation of a medical data analysis and anomaly detection system for medical data pictures might lead to improved patient outcomes and reduced healthcare costs.

This study aims to protect patient health medical data while preserving time series, resource-intensive, safety, and confidentiality analyses. Our main contribution is as follows.

1) Implemented advanced data cleaning, handling missing values, and normalization techniques to ensure consistent, high-quality, sensor-based medical data.
2) Applied discriminant principal component analysis (DPCA) to reduce data dimensionality, focusing on key features from medical images while minimizing resource consumption
3) We used recursive feature elimination (RFE) to optimize the feature selection process, ensuring that only the most relevant features are retained, enhancing the model's accuracy.
4) Introduced an auto-encoded genetic recurrent neural network (AGRNN) for accurate and efficient detection of anomalies and attacks in medical sensor data.

The remaining section of this article is organized as follows: "Literature Survey" presents the current methods literature survey. "Proposed Method" explains the suggested research methodology, comprising the protocol, algorithm, mathematical models, and pseudocode. "Experimental Results" provides simulated results by contrasting the suggested approach with current approaches. "Conclusion" summarizes the suggested approach.

## LITERATURE SURVEY

The research gaps from these earlier articles are outlined below. Consequently, deep learning-based apps for diagnosing illnesses will enable doctors to make quick clinical practice decisions and empower them in *Tsuneki (2022)*. If the training set is big and varied enough for analysis, deep learning can be more reliable for class differentiation using distinct characteristics. Still, there are enough medical photos for training sets one of the main drawbacks of deep learning in medical image analysis is that it is not always accessible from medical facilities. This review article addresses attempts and offers a few fixes for this problem to create reliable deep learning-based systems for computer-aided diagnosis that would improve clinical workflow in radiology, dentistry, endoscopy, and pathology. This research aims to provide a comprehensive overview of the literature on the use of machine learning (ML) and deep learning (DL) in the identification and categorization of various illnesses (*Rana & Bhushan, 2023*). Forty main studies from reputable publications and conferences were thoroughly investigated between January 2014 and January 2022. This summarizes several learning and deep learning methods for identifying and categorizing various illnesses, medical imaging modalities, assessment instruments and techniques, and dataset descriptions.

This effort aims to refine the prediction maps of the next epochs by utilizing knowledge from each training epoch (*Tomar et al., 2022*). To unify the prior epoch mask with the

feature map of the current training epoch, they offer a novel architecture termed feedback attention network (FANet). The learned feature maps at various convolutional layers are then given significant attention using the prior epoch mask. Moreover, the network permits iterative correction of the predictions during testing. Using seven publicly accessible biomedical imaging datasets, they demonstrate the efficacy of FANet by exhibiting a significant improvement in the majority of segmentation metrics provided by our suggested feedback attention model. This work will examine and analyze deep learning's use in picture recognition (*Li, 2022*). The development of icon recognition technology is first described in this article, which also introduces and compares the three primary deep learning models: generative adversarial networks, recurrent neural networks, and convolutional neural networks. Last, analysis and discussion of the study findings from deep learning application domains such as face identification, medical image recognition, and remote sensing image classification are conducted in *Sridhar et al. (2022)*.

Reducing discrepancies in medical imaging is essential for optimizing imaging. This article presents the GenPSOWVQ approach, a novel image compression scheme that combines wavelet VQ with a recurrent neural network. Fragments and genetic algorithms are combined to build the codebook. With reduced computing costs, the recently proposed image compression model achieves accurate compression without sacrificing picture quality when encoding clinical images. Real-time medical imaging with peak signal-to-noise ratio (PSNR), mean square error (MSE), structural similarity index (SSIM), normalised mean square error (NMSE), signal-to-noise ration (SNR), and computed radiography (CR) indicators was used to evaluate the suggested approach. The purpose of this research (*He et al., 2023*) was to increase knowledge about the uses of transformers in medical image processing. They first summarized the fundamental ideas behind the attention mechanism for transformers and other essential parts. Second, we examined the limits of specific transformer topologies designed for medical picture applications. Using transformers in various learning paradigms, increasing model efficiency, and coupling with other approaches are some of the significant difficulties they looked into for this research. The Pigeon Inspired Optimization with Encryption-based Secure Medical Image Management (PIOE-SMIM) method is presented in *Geetha et al. (2022)*.

The PIOE-SMIM technique stresses the aspects of secret share creation (SSC) and encryption procedure. First, the SSC method divides the medical images into 12 shares. The encryption technique also applies an elliptic curve cryptography (ECC) method. The PIO is employed to increase PSNR while reducing the key-making steps in the phenomenology of the mismatch model. In the end, the original images are recovered, and at the other end, they are recovered using decryption and sharing reconstruction processes. *Guan et al. (2022)* suggest using a Progressive Generative Adversarial Network for Medical Texture Image Augmentation (TMP-GAN) to enhance medical images. TMP-GAN uses the coordinated operation of looking at several channels for a generation to address the typical disadvantages of the available generation methods. Also, it uses a texture discriminative loss even more to boost the belief of a synthetic surface. In addition, TMP-GAN adopts a generating approach, similar to most production moves, to continuously enhance the accuracy of the synthesizer of medical images. To protect the images of

medical pictures as well as the privacy of the images, and at the same time promote a secure health care system, they introduce a new content-aware deoxyribonucleic acid (DNA) computing system (*Wu et al., 2022*).

Though architecture is similar in structural design, it includes different activities; the transmitter, on the one hand, and the receiver, on the other hand, do and carry out the role of performing encryption and decryption functions, respectively, in the proposed system. They also make available an arbitrary DNA binary sequence and a content-based perturbation and diffusion module for the sender and receiver. For instance, in this part of the process of DNA encoding, the above module forms a random encryption rule selector, and a few pixels from medical images are assigned to specific outputs. Still, here, the number is fixed, increasing the difficulty level of the code as a cover. Afterward, this model generates the permutation sequence that includes the redundant correlation of the neighboring pixels within the patch and the information on the pixel values. Such design increases the awareness that comes with encryption by incorporating the content of the medical picture to construct such a design, which raises the need to encounter its breakage.

These features remain consistent despite changes in scale, rotation, or illumination (*Liu et al., 2022*). This is useful in monitoring tissue movements, detecting changes, and guiding surgeons in real-time during robotic surgery or minimally invasive treatments (*Lu et al., 2023a*; *Zheng et al., 2024*). Transformers are good at capturing global dependencies because they use self-attention mechanisms that allow the model to consider relationships between all parts of the input data simultaneously (*Yin et al., 2024*; *Sun et al., 2023*). Some algorithms use machine learning models trained on large datasets of surgical images to recognize and track instruments automatically, even in complex scenes (*Lu et al., 2023b*; *Li et al., 2023*). This is particularly important in medical images, where segmentation can help identify and isolate important structures like organs, tumors, or other areas of interest (*Xia et al., 2023*; *Zhu, 2024*). A knowledge graph is built or leveraged, encodes known relationships between drugs, including existing drug-drug interactions and other relevant biological connections (*Chen et al., 2024*; *Jung et al., 2024*). Conventional ultrasound imaging often lacks the resolution to visualize these microvessels, making super-resolution techniques essential for capturing detailed images of such structures (*Luan et al., 2023*; *Yu et al., 2023*). Using synchronous multimode ultrasound offers the advantage of gathering different types of diagnostic information (structural and blood flow data) simultaneously, potentially leading to faster and more accurate diagnoses (*Yao et al., 2023*; *Qing et al., 2023*). The ML models can analyze complex interactions between variables to provide insights into the factors most predict infection (*Zhang et al., 2023*; *Lin et al., 2023*).

Some previous methods, like the AIBDA-ICH and even the PCNN, have shortcomings. AIBDA-ICH, for example, is too complex and cannot work correctly on huge amounts of medical sensor data. Similarly, PCNN is good, but the way it finds anomalies is not very resourceful regarding feature selection, leading to more resources being consumed. Support vector machine (SVM) and hidden Markov model (HMM) are unable to work on data dependence on time, which is vital in providing real-time monitoring facilities in healthcare systems. The auto-encoded genetic recurrent neural network (AGRNN) that was developed incorporates the benefits of dimensionality reduction through

autoencoders, the optimization of genetic algorithms, and the sequences of RNNs. This combined strategy enables better, more accurate, and quicker anomaly detection than the traditional AIBDA-ICH and PCNN methods, which are constrained by inadequate feature optimization and the temporal characteristics of medical sensor data.

## PROPOSED METHOD

This proposed work concentrates on revolutionary technology for big data analysis, enabling continuous health tracking and early intervention. Big data analysis and anomaly detection within sensor-based medical images are crucial for promptly identifying deviations from standard physiological patterns. This proposed work is to develop an optimized machine learning-based big data analysis and anomaly detection system tailored for sensor-based medical image data, enhancing patient care and well-being through timely alerts and interventions. Figure 1 shows the complete proposed architecture.

Combining big data and intelligent algorithmic techniques, this study offers a fresh perspective for anomaly detection in medical sensor data as it unfolds in real time. As opposed to the existing approaches wherein every concern is dealt with differently, our framework extends the concept of feature extraction, feature selection, and whole security in a single configuration, which increases performance. Big data analytics made the extensive scaled sensor data collected from medical imaging and physiological monitoring more manageable and processed. In the pre-processing stage, data of various forms and types was organized, and big data techniques such as restructuring, cleaning, and sourcing in unstructured forms were used. In the feature extraction stage, data dimensionality reduction was accomplished using DPCA, while RFE was applied to select data features and optimize the system for anomaly detection. The last step in improving accuracy in the system was when the AGRNN model used these large databases, which had a high volume of real-time anomaly detection features that offered high efficiency. There are three crucial components in the AGRNN that have been highlighted in the present study: an autoencoder, a genetic optimization layer, and a recurrent neural network.

The autoencoder compresses the input data by encoding pertinent features, which are subsequently filtered through a genetic algorithm layer for the purpose of optimizing feature selection. After that, a recurrent neural network with gated recurrent units (GRU) is used on this sequential data in order to capture the temporal relations needed for anomaly detection in real-time, close to real-time, due to the nature of the data-real-time medical sensor data. This hybrid structure allows the system to accomplish anomaly detection tasks better and with less resource consumption than the traditional CNN-based approaches such as PCNN.

### Data collection and pre-processing

The data incorporated in the proposed system were obtained from diverse medical sensor units which capture vital parameters such as heart rate, pulse rate, blood pressure, oxygen saturation, and several other physiological parameters. These sensors generate continuous streams of data from patients in different healthcare environments. The collected data, including regular and abnormal patterns, were pre-processed to remove noise, handle

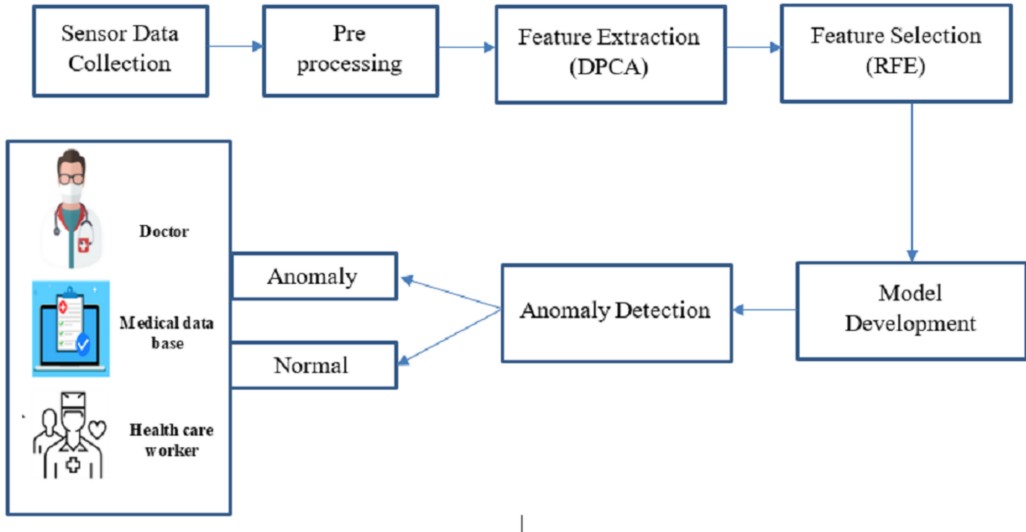

**Figure 1 Overall architecture of this research.**

**Table 1 Sample collected data.**

| S.no | Heart rate (bpm) | Pulse rate (bpm) | Blood pressure (mmHg) | Oxygen saturation (%) | Respiration rate (breaths/min) |
|------|------------------|------------------|-----------------------|-----------------------|--------------------------------|
| 1 | 72 | 72 | 120/80 | 98 | 18 |
| 2 | 75 | 74 | 122/81 | 97 | 17 |
| 3 | 73 | 73 | 121/79 | 98 | 18 |
| 4 | 74 | 73 | 123/80 | 99 | 19 |
| 5 | 76 | 75 | 124/82 | 97 | 18 |

missing values, and ensure consistency for further analysis. Using machine learning techniques, the sensor data were structured into usable formats suitable for feature extraction, selection, and anomaly detection. Table 1 shows the data collected from the sample.

### Kalman filtering

The theory behind the Kalman filter implies that a system on the form generates the signal $\mathscr{X}_{\mathscr{M}}$ that has to be filtered.

$$\mathscr{Y}_{\mathscr{M}+1} = f(\mathscr{Y}_{\mathscr{M}}, \mathscr{V}_{\mathscr{M}}) + \mathscr{U}_{\mathscr{M}} \tag{1}$$

$$\mathscr{X}_{\mathscr{M}} = \mathscr{G}(\mathscr{Y}_{\mathscr{M}}, \mathscr{V}_{\mathscr{M}}) + \mathscr{Z}_{\mathscr{M}} \tag{2}$$

In this case, $\mathscr{X}$ is the measured output, $\mathscr{V}$ is the system's input, $\mathscr{U}$ is the process noise, and $\mathscr{Z}$ is the measurement noise. It is assumed that these noise processes are white, zero-mean Gaussian processes and that their covariance is provided by the matrices Q and R, in that order:

$$\mathcal{U}_\mathcal{M} \sim \mathcal{F}(0, \mathcal{P}) \mathcal{Z}_\mathcal{M} \sim \mathcal{F}(0, \mathcal{S}) \tag{3}$$

With every time step, the Kalman filter calculates the state estimate $\widehat{\mathcal{Y}}$ and the state covariance matrix $\widehat{\mathcal{Q}}$. A step-ahead prediction of the state is made using the system model in Eq. (1). $\widehat{\mathcal{Y}}$ and $\widehat{\mathcal{Q}}$ represent the a priori estimate that is produced as a result of this, which is known as the time update. This is followed by the measurement update. The a posteriori estimates $\overline{\mathcal{Y}}$ and $\overline{\mathcal{Q}}$ is therefore produced by updating the a priori estimate with a measurement that has known measurement noise. It is typical for measurements to be made infrequently or at irregular intervals in glucose data collections. Each measurement update is handled by the filter by performing several time updates. The difference between the a posteriori and a posteriori estimates is the same in time steps when no measurement is available. Here are the formulas for the Kalman filter:

$$\overline{\mathcal{Y}}_\mathcal{M} = \Phi_{\mathcal{M}-1}\widehat{\mathcal{Y}}_{\mathcal{M}-1} + \mathcal{C}_{\mathcal{M}-1}\mathcal{V}_{\mathcal{M}-1} \tag{4}$$

$$\overline{\mathcal{Q}}_\mathcal{M} = \Phi_{\mathcal{M}-1}\widehat{\mathcal{Q}}_{\mathcal{M}-1}\Phi_{\mathcal{M}-1}^\mathcal{N} + \mathcal{P}_{\mathcal{M}-1} \tag{5}$$

$$\mathcal{A}_\mathcal{M} = \overline{\mathcal{Q}}_\mathcal{M}\mathcal{K}_\mathcal{M}^\mathcal{N}(\mathcal{K}_\mathcal{M}\overline{\mathcal{Q}}_\mathcal{M}\mathcal{K}_\mathcal{M}^\mathcal{N} + \mathcal{S}_\mathcal{M})^{-1} \tag{6}$$

$$\widehat{\mathcal{Y}}_\mathcal{M} = \mathcal{A}_\mathcal{M}(\mathcal{X}_\mathcal{M} - \mathcal{K}_\mathcal{M}\overline{\mathcal{Y}}_\mathcal{M}) \tag{7}$$

$$\widehat{\mathcal{Q}}_\mathcal{M} = (\mathcal{I} - \mathcal{A}_\mathcal{M}\mathcal{K}_\mathcal{M})\overline{\mathcal{Q}}_\mathcal{M} \tag{8}$$

$\mathcal{K}$ is the measurement matrix, and $\Phi$ is the state transition matrix. These matrices may generally be determined by linearizing $f$ and $\mathcal{G}$ in the measurement equations, provided that the system and/or the measurement equations are nonlinear.

### Smoothing

The above-discussed Kalman filter is appropriate for real-time data processing since it only estimates based on historical data. Smoothing may further enhance the estimations in an offline environment when the whole data set is accessible. Smoothing uses all the data available before and after the estimate at time k to get the estimate at time k. This is achieved using the Rauch-Tung-Striebel (RTS) algorithm. The sequences of *a priori* and *a posteriori* estimates $\overline{\mathcal{Y}}_\mathcal{M}$, $\widehat{\mathcal{Y}}_\mathcal{M}$, and state covariance matrices $\overline{\mathcal{Q}}_\mathcal{M}$ *and* $\widehat{\mathcal{Q}}_\mathcal{M}$ are stored by RTS after it first performs a forward pass of the data using the conventional Kalman filter (Eqs. (4)–(6)), these are then fed into a backward pass, which yields the following smoothed estimations, $\widehat{\mathcal{Y}}_\mathcal{M}^\mathcal{R}$ *and* $\widehat{\mathcal{Q}}_\mathcal{M}^\mathcal{R}$:

$$\mathcal{B}_\mathcal{M} = \widehat{\mathcal{Q}}_\mathcal{M}\Phi_\mathcal{M}\overline{\mathcal{Q}}_{\mathcal{M}+1}^{-1} \tag{9}$$

$$\widehat{\mathcal{Y}}_\mathcal{M}^\mathcal{R} = \widehat{\mathcal{Y}}_\mathcal{M} + \mathcal{B}_\mathcal{M}(\widehat{\mathcal{Y}}_{\mathcal{M}+1}^\mathcal{R} - \overline{\mathcal{Y}}_{\mathcal{M}+1}) \tag{10}$$

$$\widehat{\mathcal{Q}}_\mathcal{M}^\mathcal{R} = \widehat{\mathcal{Q}}_\mathcal{M} + \mathcal{B}_\mathcal{M}(\widehat{\mathcal{Q}}_{\mathcal{M}+1}^\mathcal{R} - \overline{\mathcal{Q}}_{\mathcal{M}+1})\mathcal{B}_\mathcal{M}^\mathcal{N} \tag{11}$$

By using two standard deviations (SD) to approximate a 95% confidence interval under the Gaussian assumption, the error bands, as well as the remainder of the article, are built. Within the $\widehat{\mathcal{Q}}_\mathcal{M}^\mathcal{R}$ or $\backslash\widehat{\mathcal{Q}}_\mathcal{M}$ covariance matrix, the diagonal element corresponding to the glucose state is represented by the square root, which is the SD. Every result shown here has been smoothed using fixed-interval smoothing, which uses all the accessible data. That

is, the interval is the whole set of data. It might also be feasible to create a fixed-lag smoother, which is intriguing in a situation where data is estimated inside a predetermined frame and is useful in almost real-time scenarios. When using the smoother's interpolated values for further analysis, it makes sense, and when the estimates are too erratic to be used, one might analyze the error band. For instance, a threshold for the highest permitted estimated error band might be used to do this.

### Normalization

Normalization, also known as contrast stretching, is modifying the range of pixel intensity values. This procedure is a quick and important pre-processing step that enhances the image quality by removing noise. Since each pixel's intensity must be changed for the data to be normalized, the entire data must be changed to one of the predetermined values. Normalization is a pixel-by-pixel process that preserves the ridge and valley structure's contrast and clarity. The following is the definition of the normalized data $K(\rho, v)$:

$$K(\rho, v) = \begin{cases} \mathcal{N}_0 + \sqrt{\frac{VAR_0(\mathcal{H}(\rho,v) - \mathcal{N})^2}{VAR}}, & if \quad \mathcal{H}(\rho, v) > N \\ \mathcal{N}_0 - \sqrt{\frac{VAR_0(\mathcal{H}(\rho,v) - \mathcal{N})^2}{VAR}} \end{cases} \qquad (12)$$

where $\mathcal{N}_0$ and $VAR_0$, which are respectively the intended (pre-specified) mean and variance values, are defined as follows:

$$\mathcal{N}(\mathcal{H}) = \frac{1}{\mathcal{M}^2} \sum_{\rho=0}^{\mathcal{M}-1} \sum_{v=0}^{\mathcal{M}-1} \mathcal{H}(\rho, v), VAR(\mathcal{H}) = \frac{1}{\mathcal{M}^2} \sum_{\rho=0}^{\mathcal{M}-1} \sum_{v=0}^{\mathcal{M}-1} \mathcal{H}(\rho, v) - \mathcal{N}(\mathcal{H}))^2 \qquad (13)$$

To make the calculations that follow easier, we have set $\mathcal{N}_0 = 0$ and $VAR_0 = 1$ in this work. This ensures that the new pixel intensities for the normalized data will primarily fall between $-1$ $and$ $1$.

## Feature extraction

In this research work, feature extraction technology was adopted that uses the discriminant principle component analysis (DPCA) algorithm. DPCA is a mixed method of principal component analysis PCA and linear discriminant analysis LDA techniques. In WBANs, feature extraction uses DPCA to reduce the amount of data that needs to be transmitted by the sensors which in turn helps to conserve resources and energy. DPCA enables a further reduction in the dimensionality of the extracted components. Supervised DPCA aims to identify linear feature combinations that optimize the distance between classes. In the feature extraction method, we extract features such as heart rate images, respiration rate images, blood pressure images, oxygen saturation images, electromyography (EMG) images, skin temperature images, ambient temperature, muscle activity, and cross-correlation.

### Principle component analysis

A method for reducing linear dimensionality that projects data in the direction of the greatest variation is principle component analysis (PCA). The ECG data collection is

processed using this approach to extract pertinent information by $\mathscr{X}(\mathscr{M})$ as in Eq. (14). This represents the signal segment of a heartbeat.

$$\mathscr{X}(\mathscr{M}) = \begin{bmatrix} \mathscr{X}(1) \\ \mathscr{X}(2) \\ \vdots \\ \mathscr{X}(\mathscr{O}) \end{bmatrix} \tag{14}$$

where $\mathscr{O}$ is the number of heartbeat samples collected. Equation (15) states that the heartbeats $\mathscr{X}_1, \mathscr{X}_2, \ldots..\mathscr{X}_{\mathscr{T}}$ are thus $\mathscr{T}$ observations of heartbeats. The matrix $\mathscr{O} \times \mathscr{T}$ depicts the whole collection of heartbeats.

$$\mathscr{X} = [\mathscr{X}_1 \mathscr{X}_2 \ldots..\mathscr{X}_{\mathscr{T}}] \tag{15}$$

The stages involved in the PCA are as follows:

1. Find the mean vector using computation. Every heartbeat's mean vector is computed using the formula in Eq. (16).

$$\overline{\mathscr{X}} = \frac{1}{\mathscr{O}} \sum_{j=1}^{\mathscr{O}} \mathscr{X}_j \tag{16}$$

2. Use Eqs. (17) and (18) to calculate the mean adjusted values.

$$\mathscr{X} ad\, j_j = \mathscr{X}_j - \overline{\mathscr{X}} \tag{17}$$

$$\mathscr{X} ad\, j_j = [\mathscr{X} ad\, j_1 \quad \mathscr{X} ad\, j_2 \quad \ldots \quad \mathscr{X} ad\, j_{\mathscr{T}}] \tag{18}$$

3. As shown by Eq. (19), compute the covariance matrix.

$$\mathscr{B} = \frac{1}{\mathscr{O} - 1} \sum_{j=1}^{\mathscr{O}} (\mathscr{X}_j - \overline{\mathscr{X}})^{\mathscr{N}} (\mathscr{X}_j - \overline{\mathscr{X}}) \tag{19}$$

4. The covariance matrix's eigenvalues and eigenvectors should be determined. The eigenvalues $\alpha_j$ and $d_j$ eigenvectors line up with Eq. (20).

$$\mathscr{B}.d_j = \alpha_j.d_j \quad j = 1, \ldots..\mathscr{T} \tag{20}$$

5. Forming a feature vector by selecting components. The primary component is the eigenvector with the largest value. Subsequently, the components are retrieved in priority order based on the eigenvectors' order of eigenvalues from highest to lowest. Choosing K-principal components that preserve the physiological information subsequently reduces the dimensionality. Equation (21) may be used to get the proportion of variance, $ea$ , for each eigenvalue.

$$e\mathscr{M} = \frac{\sum_{j=1}^{\mathscr{M}} \alpha_j}{\sum_{j=1}^{\mathscr{T}} \alpha_j} \tag{21}$$

6. The fresh data set's extraction. By using Eq. (22), the final dataset may be acquired.

$$\mathscr{X}_{pca(\mathscr{M})} = \hat{e} \mathscr{M}^{\mathscr{N}} \mathscr{X}_{ad\,j}{}^{\mathscr{N}} \tag{22}$$

A general technique of health monitoring multivariate data in terms of dimensionality reduction is principal component analysis in short PCA. PCA identifies and retrieves critical elements by transforming the original variables into a new set of uncorrelated variables called principle components. Health-related data analysis and comprehension become less complicated, thus enhancing pattern spotting and trend analysis for better management and diagnosis.

### Linear discriminant analysis

LDA is employed in health surveillance to improve the separation of different health states and draw out valuable features of interest. LDA focuses on the teacher classes by projecting the data on its classification hyperplane to bring out the differences. The core function of LDA and its application in health monitoring is improving the efficiencies of the predictive and diagnostic tasks by determining the defining characteristics of the particular medical diagnosis.

The low-dimensional complement space of $\mathscr{R}_c$ null space, related to $\mathscr{R}'$, is first extracted. Let $\mathscr{R}_c$ and $\mathscr{R}_z$ be among and within-class scatter matrices. Let $\mathscr{U}_c = [\mathscr{U}_{c1}, \ldots \mathscr{U}_{c\mathcal{O}}]$ be the $\mathcal{O}$ eigenvectors of $\mathscr{R}_c$ that correspond to the $\mathcal{O}$ non-zero eigenvalues $\mathscr{D} = [\alpha_{c1}, \ldots, \alpha_c\mathcal{O}]$, where $\mathcal{O} = \min(\mathscr{B} - 1, \mathscr{I})$.

$\mathscr{U}_c$ spans the $\mathscr{R}_c$ subspace $\beta'$ and is further scaled by $\mathscr{V} = \mathscr{U}_c \mathscr{D}_c^{-1/2}$ to get $\mathscr{V}^{\mathscr{N}} \mathscr{R}_c \mathscr{V} = \mathscr{J}$, where $\mathscr{J}$ is the diagonalization operator and $\mathscr{D}_c = diga\,(\mathscr{D}), diag()$ indicates the diagonalization operator. $\mathcal{O} \times \mathcal{O}$ identity matrix using Eq. (23):

$$\sum{}'_j(\gamma, \rho) = (1 - \rho) \sum{}'_j(\gamma) + \frac{\rho}{\mathcal{O}} tr\left[\sum{}'_j(\gamma)\right] \mathscr{J},$$

$$\sum{}'_j(\gamma) = \frac{1}{\mathscr{B}_j(\gamma)}\left[(1 - \gamma)\mathscr{R}_j + \gamma\mathscr{R}\right], \tag{23}$$

$\mathcal{O}$ is the dimensionality of $\beta'. \mathscr{B}_j(\gamma) = (1 - \gamma)\mathscr{B}_j + \gamma\mathscr{F}$, and $\mathscr{R}_j$ is the covariance matrix of the i$^{th}$ class evaluated in $\beta'$, that is, $\mathscr{R}_j = \sum_{i=1}^{\mathscr{B}_j} (\mathscr{X}_{ji} - \overline{\mathscr{X}}_j)(\mathscr{X}_{ji} - \overline{\mathscr{X}}_j)^{\mathscr{N}}$, $\mathscr{X}_{ji} = \mathscr{V}^{\mathscr{N}}\mathscr{W}_{ji}, \overline{\mathscr{X}}_j = (1/\mathscr{B}_j)\sum_{i=1}^{\mathscr{B}_j}\mathscr{X}_{ji}$ and $\mathscr{R} = \sum_{j=1}^{\mathscr{B}}\mathscr{R}_j$.

In kernel space $\mathbb{G}^g$, let $\Phi = [\phi(\mathscr{W}_{11}), \ldots, \phi(\mathscr{W}_{\mathscr{B}\mathscr{B}b})]$ represent the appropriate feature representations of the training samples. Assume that $\mathscr{M}$ is an $\mathscr{F} \times \mathscr{F}$ Gram matrix, that is, $\mathscr{M} = (\mathscr{M}_{\ell g})_{\ell=1,\mathscr{B}}^{\mathcal{G}=1,\mathscr{B}_{\mathcal{J}g}}$ is a $\mathscr{B}_\ell \times \mathscr{B}_{\mathcal{G}}$ sub-matrix of $\mathscr{M}$ made up of samples from classes $\mathscr{I}_1$ and $\mathscr{W}_g$, that is, $\mathscr{M}_{\ell g} = (\mathscr{M}_{ji})_{j=1,\ldots,\mathscr{B}_1}^{i=1,}$, where $\mathscr{M}_{ji} = \mathscr{M}(\mathscr{W}_{\ell j}, \mathscr{W}_{\mathcal{G}i})$ and $\mathscr{M}(.)$ denotes the kernel function described in $\mathbb{S}^{\mathscr{I}}. \overline{\mathscr{R}}_c$, which is defined as the between-class scatter in $\mathbb{G}^g$ by Eq (24),

$$\acute{\mathscr{R}}_c = \frac{1}{\mathscr{F}}\sum_{j=1}^{\mathscr{B}}\mathscr{B}_j(\overleftarrow{\phi}_j - \overleftarrow{\phi})(\overleftarrow{\phi}_j - \overleftarrow{\phi})^{\mathscr{N}} \tag{24}$$

where the training sample $\mathscr{G}\mathscr{G}$ mean $\overline{\phi}_j = (1/\mathscr{B}_j)\sum_{j=1}^{b}\phi(\mathscr{W}_{jı})$ and the $\mathscr{X}_j$ mean in $\mathbb{G}\mathscr{G}$ is $\overline{\phi} = \frac{1}{\mathscr{F}}\sum_{j=1}^{\mathscr{B}}\sum_{ı=1}^{\mathscr{B}_j}\phi(\mathscr{W}_{jı})$. $\grave{\mathscr{U}}_c = [\overline{\mathscr{U}}_{c1},\ldots,\overline{\mathscr{U}}_{c\mathcal{O}}]$, the eigenvectors of $\mathscr{R}_c$ that correspond to $\mathcal{O}$ the biggest eigenvalues. The value of $\grave{\mathscr{U}}_c$ is derived by resolving the eigenvalue problem of $\overline{\mathscr{R}}_c$, denoted by Eq. (25):

$$\overline{\mathscr{R}}_c = \sum_{j=1}^{b}\left(\sqrt{\frac{bj}{\mathscr{F}}}(\overline{\phi}_j - \overline{\phi})\right)\left(\sqrt{\frac{bj}{\mathscr{F}}}(\overline{\phi}_j - \overline{\phi})\right)^{\mathscr{N}} = \sum_{j=1}^{b}\grave{\phi}_j\grave{\phi}_j^{\mathscr{N}} = \Phi_c\Phi_c^{\mathscr{N}} \tag{25}$$

where $\dot{\phi}_j = \sqrt{\mathscr{B}_j/\mathscr{F}}\left(\overline{\phi}_j - \overline{\phi}\right)$ and $\Phi_c = [\dot{\phi}_1,\ldots,\dot{\phi}_c]$. A matrix of size $\mathscr{G}\times\mathscr{G}$, where $\mathscr{G}$ is the dimensionality of kernel space, is provided for $\mathscr{R}_c$. It is not feasible to calculate the eigenvectors of $\overline{\mathscr{R}}_c$ directly because of the $\mathscr{K}\mathscr{H}$ of $\mathbb{G}^{\mathscr{G}}$. $\left(\Phi_c\Phi_c^{\mathscr{N}}\right)(\Phi_c\bar{n}_{cj}) = \overline{\gamma}_{cj}(\Phi_c\bar{n}_{cj})$. As a result, it may be concluded that $\Phi_c\bar{n}_{cj}$. As an example, the $j^{th}$ eigenvector of

$$\overline{\mathscr{R}}_c = \Phi_c\Phi_c^{\mathscr{N}} - \Phi_c^{\mathscr{N}}\Phi_c = \tag{26}$$

$$\frac{1}{\mathscr{F}}\mathscr{C}\left(\mathscr{D}_{\mathscr{F}\mathscr{B}}^{\mathscr{N}}.\mathscr{A}.\mathscr{D}_{\mathscr{F}\mathscr{B}} - \frac{1}{\mathscr{F}}\left(\mathscr{D}_{\mathscr{F}\mathscr{B}}^{\mathscr{N}}.\mathscr{A}.1_{\mathscr{F}\mathscr{B}}\right) - \frac{1}{\mathscr{F}}\left(1_{\mathscr{F}\mathscr{B}}^{\mathscr{N}}.\mathscr{A}.\mathscr{D}_{\mathscr{F}\mathscr{B}}\right) + \frac{1}{\mathscr{F}^2}\left(1_{\mathscr{F}\mathscr{B}}^{\mathscr{N}}.\mathscr{A}.1_{\mathscr{F}\mathscr{B}}\right)\right)$$

where $\mathscr{C} = diag\left[\sqrt{\mathscr{B}_1},\ldots,\sqrt{\mathscr{B}_\mathscr{B}}\right]$ is a $\mathscr{F}\times\mathscr{B}$ block diagonal matrix and a $\mathscr{B}_j$ is a $\mathscr{B}_j = 1$ vector with all members equal to $1$, $\mathscr{D}_{\mathscr{F}\mathscr{B}} = diag[b_{\mathscr{B}_1},..,b_{\mathscr{B}_\mathscr{B}}]$ matrix with all elements equal to $1/\mathscr{B}_j$ Assume that $\overline{\mathfrak{C}}_{c\mathcal{O}} = [\bar{n}_{c1},,\grave{n}_{c\mathcal{O}}]$. It is easy to deduce that $\overline{\mathscr{U}}_c^{\mathscr{N}}\overline{\mathscr{S}}_c\overline{\mathscr{U}}_c = \overline{\wedge}_c$, where $\overline{\wedge}_c = diag\left[\overline{\gamma}_{c,1}^2,\ldots,\overline{\gamma}_{c,\mathcal{O}}^2\right]$, is made up of $\mathcal{O}$ significant eigenvectors of $\Phi_c^{\mathscr{N}}\Phi_c$, which correspond to $\mathcal{O}$ biggest eigenvalues $\overline{\gamma}_{c1}>,\ldots,>\overline{\gamma}_c$ and $\grave{\mathscr{U}}_c = \Phi_c\overline{\mathfrak{C}}_{c\mathcal{O}}$. Consequently, the transformation matrix $\overrightarrow{\mathscr{V}}$ Eqs. (27), (28) evaluates $\overrightarrow{\mathscr{V}}$ so that $\overrightarrow{\mathscr{V}}^{\mathscr{N}}\mathscr{R}_c\overrightarrow{\mathscr{V}}$ is evaluated.

$$\overline{\mathscr{V}} = \overline{\mathscr{U}}_c\overline{\mathscr{D}}_c^{-1/2}, \overline{\mathscr{U}}_c = \Phi_c\grave{\mathfrak{C}}_{c\mathcal{O}} \tag{27}$$

$$\grave{\mathscr{X}}_{jı} = \overline{\mathscr{V}}^{\mathscr{N}}\phi(\mathscr{W}_{jı}) = \overline{\mathscr{D}}_c^{-1/2}\mathfrak{C}_{c\mathcal{O}}^{\mathscr{N}}\Phi_c^{\mathscr{N}}\phi(\mathscr{W}_{jı}) \tag{28}$$

where $\Phi_c^{\mathscr{N}}\phi(\mathscr{W}_{jı})$ may be written as Eq. (29).

$$\Phi_c^{\mathscr{N}}\phi(\mathscr{W}_{jı}) = \frac{1}{\sqrt{\mathscr{F}}}\mathscr{C}\left(\mathscr{D}_{\mathscr{F}\mathscr{B}}.u(\phi(\mathscr{W}_{jı})) - \frac{1}{\mathscr{F}}1_{\mathscr{F}\mathscr{B}}^{\mathscr{N}}.u(\phi(\mathscr{W}_{jı}))\right) \tag{29}$$

The kernel function outlined in $\mathbb{S}^{\mathscr{I}}$, $\phi(\mathscr{W}_{oe})\phi(\mathscr{W}_{jı,}) = \mathscr{A}(\mathscr{W}_{oe},\mathscr{W}_{jı})$, is used to evaluate $u(\phi(\mathscr{W}_{jı}) = [\phi(\mathscr{W}_{11}\phi(\mathscr{W}_{jı}), \phi(\mathscr{W}_{12}\phi(\mathscr{W}_{jı}),\ldots,\phi(\mathscr{W}_{\mathscr{B}\mathscr{B}_\mathscr{B}}\phi(\mathscr{W}_{jı}))]^{\mathscr{N}}$.

$$\sum_j \grave{}(\beta,\rho) = (1-\rho)\sum_j \grave{}(\beta) + \frac{\rho}{\mathcal{O}}tr\left[\sum_{j=1}^{}\beta\right]\mathscr{I},$$

$$\sum_j \grave{}(\beta) = \frac{1}{\mathscr{B}_j(\beta)}\left[(1-\beta)\overline{\mathscr{R}}_j + \beta\overline{\mathscr{R}}\right],$$

$$\grave{\mathscr{R}}_j = \sum_{ı=1}^{\mathscr{B}_j}(\overline{\mathscr{X}}_{jı} - \overline{\overline{\mathscr{X}}}_j)(\overline{\mathscr{X}}_{jı} - \overline{\overline{\mathscr{X}}}_j)^{\mathscr{N}},$$

$$\overline{\mathscr{R}} = \sum_{j=1}^{\mathscr{B}}\grave{\mathscr{R}}_j.$$

$$\overline{\overline{\mathcal{X}}}_j = \left(\frac{1}{\mathcal{B}_j}\right) \sum_{i=1}^{\mathcal{B}_j} \overline{\mathcal{X}}_{ji} \tag{30}$$

The regularization parameters are denoted by $(\beta, \rho)$. Equation (31) provides the covariance matrix of the $j^{th}$ class, or $\overline{\mathcal{R}}_j$, which is the crucial element in the assessment of $\sum_j (\beta, \rho)$.

$$\grave{\mathcal{R}}_j = \sum_{i1}^{\mathcal{B}_j} (\grave{\mathcal{X}}_{ji} - \overline{\overline{\mathcal{X}}}_j)(\overline{\mathcal{X}}_{ji} - \overline{\overline{\mathcal{X}}}_j)^{\mathcal{N}} = \sum_{i1}^{\mathcal{B}_j} \left(\overline{\mathcal{X}}_{ji}\overline{\mathcal{X}}_{ji}^{\mathcal{N}}\right) -$$

$$\sum_{i1}^{\mathcal{B}_j} \left(\overline{\overline{\mathcal{X}}}_{ji}\overline{\mathcal{X}}_{ji}^{\mathcal{N}}\right) - \sum_{i1}^{\mathcal{B}_j} \left(\overline{\mathcal{X}}_{ji}\overline{\overline{\mathcal{X}}}_{i}^{\mathcal{N}}\right) + \sum_{i1}^{\mathcal{B}_j} \overline{\mathcal{X}}_{ji}\grave{\mathcal{X}}_{ji}^{\mathcal{N}} - \mathcal{B}_j\overline{\overline{\mathcal{X}}}_{ji}\overline{\mathcal{X}}_{j}^{-\mathcal{N}} - \mathcal{B}_j\overline{\overline{\mathcal{X}}}_j\overline{\overline{\mathcal{X}}}_j^{\mathcal{N}} + \mathcal{B}_j\overline{\overline{\mathcal{X}}}_j\overline{\overline{\mathcal{X}}}_j^{\mathcal{N}} =$$

$$\sum_{i1}^{\mathcal{B}_j} \grave{\mathcal{X}}_{ji}\overline{\mathcal{X}}_{ji}^{\mathcal{N}} - \mathcal{B}_j\overline{\overline{\mathcal{X}}}_j\overline{\overline{\mathcal{X}}}_j^{\mathcal{N}} = \mathcal{I}_1 - \mathcal{B}_j \times \mathcal{I}_1 \tag{31}$$

In this case, $\mathcal{I}_1 = \sum_{i=1}^{\mathcal{B}_j} \grave{\mathcal{X}}_{ji}\overline{\mathcal{X}}_{ji}^{\mathcal{N}}$ and $\mathcal{I}_2 = \overline{\overline{\mathcal{X}}}_j\overline{\overline{\mathcal{X}}}_j^{\mathcal{N}}$.

To identify the test picture, the distance between each class center $\overline{\overline{\mathcal{X}}}_j$ and the feature representation of the test data $\overline{\mathcal{P}}$, or mahalanob, is employed. Specifically, $\mathcal{I}\mathcal{H}(\mathcal{Q}) = \arg min_j h_j (\overline{\mathcal{P}})$, which may be computed using Eq. (32) as follows:

$$h_j(\overline{\mathcal{P}}) = \left(\overline{\mathcal{P}} - \overline{\overline{\mathcal{X}}}_j\right)^{\mathcal{N}} \sum_j{}^{-1}(\beta, \rho)\left(\overline{\mathcal{P}} - \overline{\overline{\mathcal{X}}}_j\right) + ln\left|\sum_j(\beta, \rho)\right| - 2ln\varphi_j \tag{32}$$

where $\varphi_j = \mathcal{B}_j / \mathcal{F}$.

$$\left(\grave{\mathcal{D}} = \arg max_{\mathcal{D}} \quad -\frac{\left|\grave{\mathcal{D}}^{\mathcal{N}} \grave{\mathcal{R}}_c \grave{\mathcal{D}}\right|}{\left|\grave{\mathcal{D}}^{\mathcal{N}} \overline{\mathcal{R}}_c \overline{\mathcal{D}}\right|} + \left|\grave{\mathcal{D}}^{\mathcal{N}} \grave{\mathcal{R}}_z \grave{\mathcal{D}}\right|\right)$$

when $\left(\beta = 1, \rho = \left(tr\left(\frac{\overline{\mathcal{R}}_i}{\mathcal{F}}\right) + \mathcal{O}\right)/\mathcal{O}\right)$.

By helping to discover distinguishing characteristics that help differentiate various medical disorders, LDA enhances the effectiveness of monitoring and diagnostic processes in the health sector.

## Feature selection

In this research, we used the recursive feature elimination (RFE) algorithm for feature selection. RFE is a feature selection method that periodically reduces the dataset's most minor significant features and helps avoid overfitting. It operates by repeatedly deleting the feature with the lowest relevance score after training the model on the whole collection of features; after each feature removal, the model is retrained, and this process continues until the required amount of features is obtained in Algorithm 1. In the feature selection method, we select features like EMG, heart rate images, and blood pressure images.

| **Algorithm 1** Representation of procedure for RFE method. |
| :--- |
| 1: **Input:** Dimensionality diminished data |
| 2: **Output:** Select significant features (eliminate unessential features) |
| 3: Implement cross-validation to train the classification model using all of the features. |
| 4: Determine model performance. |
| 5: Determine the feature's ranking or relevance. |
| 6: **for** each subset $T_j$ with $j = 0, 1, 2, 3, \ldots, e$ **do** |
| 7:     Preserve the most essential $T_i$ attributes. |
| 8:     Use Eq. (20) to update the adaptive learning function stage. |
| 9:     Recalculate the model's output. |
| 10:    Recalculate each feature's rating importance. |
| 11: **end for** |
| 12: Establish the ideal feature count. |

Therefore, the main goal of this study is to develop a new approach (called RFE) that modifies the standards for eliminating characteristics from each state.

RFE with an adaptive learning function basis. This is where the adaptive learning function is shown. Consequently, it significantly improves generalization accuracy, especially for small numbers of features.

$$|\mathscr{Z}_\iota| = \left| \sum_{j=1}^{e} \beta_j \mathscr{X}_j \mathscr{Y}_{j\iota} \right|, \tag{33}$$

In where $\mathscr{Y}_{j\iota}$ represents the feature vector's $\iota^{th}$ member ($j^{th}$).

The properties of $\mathscr{Q}$ may be rated according to relevance (a higher value indicates greater significance) after the computation of Eq. (33). The weights were combined using an adaptive learning function $\phi$, which had the following definition:

$$\phi\left(\mathscr{Z}_{j\iota}, d_j\right) = \mathscr{Z}_j \frac{1}{rank(d_j)}, \tag{34}$$

$$Rank(d_j) = |d_k| \big| d_k \geq d_j \big|, \tag{35}$$

where $d_j$ is the resultant attribute rank for the $j^{th}$ feature in Eq. (34) and $\mathscr{Z}_j$ is the RFE weight produced by Eq. (33). The findings' initial rankings were converted into a rank-based form using the rank function instead of utilizing the original ranks (Eq. (35)). This suggests that features with the highest rank will be assigned a weight of 1, followed by features with the second-highest rank being assigned a weight of 2, features with the third-highest rank being assigned a weight of 3, and so on, up to $\mathscr{Q}$.

## Attack detection using auto-encoded genetic recurrent neural network (AGRNN)

In this research, we used an auto-encoded genetic recurrent neural network (AGRNN) algorithm for attack detection. The AGRNN may be utilized to predict the next data point in a time series after it has been trained on typical data. Comparing what was predicted to the actual observed value is a common way to find anomalies. An anomaly may be present if the discrepancy between the expected and observed values exceeds a certain threshold.

### Autoencoder

A kind of neural network called an autoencoder has many layers, and its goal output is the same as the input with a small degree of reconstruction error since the output is less altered and more like the input. The autoencoder encodes the input and then uses unsupervised learning to decode or reconstruct the output. To minimize the number of dimensions in features, extract pertinent features, eliminate noise from data, forecast sequences, and identify abnormalities, autoencoders are often used in recommender systems. To keep things brief, we will discuss the overall architecture of an autoencoder without getting into specifics.

Encoder, bottleneck, decoder, and reconstruction loss are the four main parts of a generic autoencoder in Fig. 2. The encoder compresses the input data into an encoded form while helping the model decrease the input characteristics. The "bottleneck" layer is the one that has the least amount of features and is composed of compressed input data. The decoder aids the model in reconstructing the output from the encoded representation and ensures that the output and input are the same.

Reconstruction loss is the last step, which involves assessing the decoder's performance and calculating how close the output is to the original input. Backpropagation is included to carry out training and reduce reconstruction loss even further. This minimal loss illustrates the objective that AE attempts to accomplish. The input $\mathscr{Y}$ will be compressed using the encoder function $\mathfrak{E}$ and stored in $\mathscr{W} = \mathfrak{E}(\mathscr{Y}), \mathscr{Y}' = \mathscr{H}(\mathfrak{E}(\mathscr{Y}))$ is how the Decoder $\mathscr{H}$ will attempt to replicate the input. The difference between the encoded and decoded vectors, in this case, is the reconstruction loss.

One technique for calculating the reconstruction loss is the mean square error, or MSE. It's provided by $\mathscr{U}$.

$$loss(\mathfrak{E}, \mathscr{H}) = \frac{1}{e} \sum_{j=1}^{e} (\mathscr{Y}^j - \mathscr{H}(\mathfrak{E}(\mathscr{Y}^j)))^2$$

Variational autoencoders (VAEs) employ Kullback-Leibler (KL) divergence, an additional technique for computing reconstruction loss. KK One way to quantify the differences between two probability distributions, the probability distribution of the data in the latent space and the probability distribution of the data being projected into the latent space, is to use the non-negative number known as divergence. Variational autoencoders (VAEs) employ KL divergence, an additional technique for computing reconstruction loss.

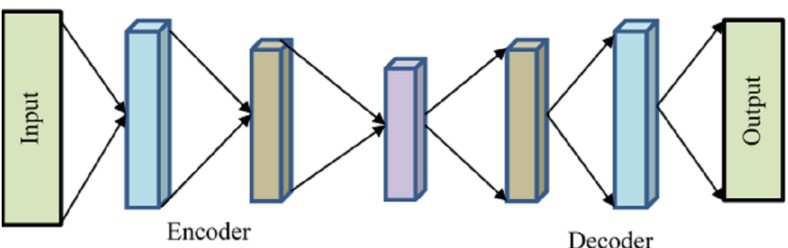

**Figure 2 Autoencoder.**

KK The difference between two probability distributions, the probability distribution of the data in the latent space and the probability distribution of the data being projected into the latent space, is measured by the non-negative number known as divergence.

### Genetic algorithm

Anomaly detection is improved using the genetic algorithm in health data while optimally selecting and combining features to enhance the efficiency of the intrusion detection model. This evolutionary strategy consists of a few basic steps. The first is the crossover, which combines two or more solutions by taking parts of their genetic material. Usually, two parents are required to contribute to the offspring's birth, although occasionally, this is accomplished by a single parent. It is customary to carry out P-point crossover. In this manner, the elements of the solutions are sequenced, cut at several specified points, and rearranged. The next step is the mutation step, which randomly modifies the solutions. The mutation rate specifies the magnitude of such changes to allow mobility to all areas of the solution space (reachability), avoid introducing any one-sidedness (impartiality), and make the degree of mutation controllable (scalability). The new population created is subjected to genotype-phenotype mapping following crossover and mutation. The new offspring population is assessed by how it looks, the phenotype, to guarantee that no favorable conditions affect the process. The quality of every solution is evaluated during the first step of the process, called the fitness step, using a fitness function to check how well the particular problem of the algorithm. In the selection phase, there is a phenomenon of surviving selection, which assists in selecting the parents that allow their genes to be carried to the next generation, which aligns with Darwin's principle of survival. This selection procedure also determines which of the parents is to be included for contribution in the generation of the next individual during the crossover process. Lastly, the termination step stipulates how the evolutionary cycle is terminated, or the last generation is reached (imposed by other restrictions such as the time or the cost of fitness), deciding upon a preset number of generations. Throughout these steps, the genetic algorithm repeatedly improves the model that detects anomalous behavior from the medical data by applying the genetic method to improve feature search and selection and enhance detection performance.

The pseudocode for the GA algorithm is given by Algorithm 2.

**Algorithm 2** GOA algorithm.

1: **Input:** Random generated key

2: **Output:** Producing optimal key

3: Initialize: let the random distribution of population parameters $P_i$ and their values of $c_{max}$, $c_{min}$, and $L$ be determined.

4: Calculate the fitness measure, $P_j$, for each agent that will perform a search.

5: $T$ is seen as the most effective search agent.

6: **while** $t < L$ **do**

7:     do $c$.

8:     **for** each search agent **do**

9:         Disturb the position of $k$ grasshoppers, such that the distance is stabilized within 1–4 units.

10:         Move the current search agent to a new position.

11:         Return the current search agent to the original position.

12:     **end for**

13:     Determine $T$ again and update.

14:     $t = t + 1$

15: **end while**

16: Return $T$ (Optimal Key).

### Recurrent neural network

This is a well-known paradigm for classifying time-related and sequential data. Here, the joining data kept in memory is the outcome of the network receiving the current value as feedback. RNN creates the predicting model at each stage, refreshes the unseen state, and receives the incoming data. RNN applies gradient descent concepts, and since its nonlinear processes are disrupted by unseen sections, it has a very dynamic nature.

A function to the current input and previous state is provided by the recurrent neural network (RNN) portion. For instance, at time $\mathfrak{T}$, the input value is shown as a; at time $\mathfrak{T} - 1$, on the other hand, the input value is $e$. A fresh stage was acquired after using the function to $b$ and $e$. The new present situation is described in Eq. (1).

$$\mathscr{G}_{\mathfrak{T}} = g(\mathscr{G}_{\mathfrak{T}-1}, \mathscr{Y}_{\mathfrak{T}}) \tag{36}$$

The new step is shown by $\mathscr{G}_{\mathfrak{T}}$ in this case, whereas the previous stage is represented by $\mathscr{G}_{\mathfrak{T}-1}$. The input value at time $\mathfrak{T}$ is shown by the variable $\mathscr{Y}_{\mathfrak{T}}$.

Next, define the $g()$ method. Tanh is a symbol for the activation process. The input value is denoted by the $\mathscr{L}_{\mathscr{Y}\mathscr{G}}$ matrix, while weights are represented by the matrix $\mathscr{L}_{\mathscr{G}\mathscr{G}}$. The equation at this point is

$$\mathscr{G}_{\mathfrak{T}} = tan\mathscr{G}(\mathscr{L}_{\mathscr{G}\mathscr{G}}\mathscr{G}_{\mathfrak{T}-1} + \mathscr{L}_{\mathscr{Y}\mathscr{G}}\mathscr{Y}_{\mathfrak{T}}) \tag{37}$$

Increase the amount of memory on the network and include further steps in the calculation, such *as* $\mathscr{G}_{\mathfrak{T}-2}$, $\mathscr{G}_{\mathfrak{T}-3}$, *etc.* Equation (38) may be used to calculate the output value throughout the testing period.

$$\mathscr{Y}_{\mathfrak{z}} = \mathscr{L}_{\mathscr{GX}}\mathscr{G}_{\mathfrak{z}} \qquad (38)$$

The output value is shown by $\mathscr{Y}_{\mathfrak{z}}$ in this case. Error data is computed by comparing the output value with an actual value.

## EXPERIMENTAL RESULTS

Using big data analysis significantly enhanced the system's ability to manage and process large sensor datasets, reducing dimensionality and computational load while improving the detection of anomalies in real-time patient monitoring. This section offers an experimental examination of the recommended optimized machine learning-based big data analysis and anomaly detection to compare it with evaluation metrics. The results show that the optimized machine learning-based big data analysis anomaly detection in sensor-based medical data analysis method that has been recommended successfully provides a strong framework that protects personal data, boosts security, and enhances patient care and well-being for responding to emergencies. This section is divided into three subsections: comparative analysis, simulation setup, and analysis.

### Experimental setup

The proposed big data analysis in sensor-based medical images using optimized machine learning methodology is implemented and simulated using NS-3.26. We tend to tune the system configurations in terms of hardware and software configurations, respectively. The hardware settings adjusted included the hard disk and RAM, which were one terabyte and eight gigabytes, respectively.

### Performance metrics

The testing model needs to measure accuracy, which is the test's ability to discriminate between sick and healthy instances. To assess the accuracy of the test, it is necessary to know the proportion of true positive (TP + TR) and true negative (TN) cases in each instance evaluated. In this case, mathematically, it is given as follows:

$$A_c = \frac{TP + TN}{TP + TN + FP + FN} \qquad (39)$$

where *FP* denoted false positive, *FN* defined the false negative. Lifetime may pertain to the total need or demand for data throughout a certain period, such as a patient's lifespan or a designated timeframe for medical research. The following might be one such connection or formula:

$$LT = DAR * TP \qquad (40)$$

Lifetime *LT* Need is the total data required for a certain time. The pace of generation or reception of fresh data is known as the data arrival rate (DAR). The term over which you want to compute the cumulative requirement is called the time period *TP*. In the context of communication or data transfer, propagation latency is the time data moves from one location to another. In the case of health data, propagation latency must be kept to a minimum to guarantee accurate and timely information transmission, particularly in

healthcare systems where real-time data might affect patient care. The following formula may be used to determine propagation latency *PL*

$$PL = \frac{D}{v} \tag{41}$$

The parameters determining the propagation velocity of *a* medium are *v*, D, and *PL*, representing propagation latency, distance, and necessary speed, respectively. The term "resource consumption" for health data describes how computer resources, including CPU, memory, storage, and network bandwidth, are used inside a system or application to process, store, and transmit data linked to health. Health data systems' functionality, scalability, and dependability require effective management of these resources.

In regression analysis, particularly the prediction of health data, the root mean square error (RMSE) is a frequently used statistic to measure the discrepancy between anticipated and actual values. RMSE can be calculated using the following formula:

$$RMSE = \sqrt{\frac{1}{R} \sum_{j=1}^{R} (_xj) - xj)2} \tag{42}$$

whereas the variable $X_j$ represents the actual value for data point *j*, *R* is the number of data points, and $X_j$ anticipated value for data point *j*. An indicator of a binary classification model's effectiveness is the true positive rate (TPR), often called sensitivity or recall. Measuring the percentage of true positive cases (such as the existence of an illness) that the model accurately detects is a common practice when analyzing health data. The formula below is used to determine the true positive rate:

$$TPR = \frac{TP}{TP + FN} \tag{43}$$

Correctly anticipated as positive cases are known as true positives *(TP)*. False negatives *(FN)* are positively anticipated but mistakenly forecasted as negative situations.

## Result analysis

A performance evaluation of the proposed system is carried out in Table 2, and it is clear that the system is superior in many determinants. It can be noted that the system can reach an accuracy value of 98%, which implies that it is very proficient in identifying various anomalies. This shows that the system can sustain its operational efficiency over time and can be measured in longevity, more or less equal to 1.1 min. Also, the propagation latency is decreased to 55.14 milliseconds, suggesting the system's responsiveness, which is essential in real-time monitoring applications.

The system resources used by the proposed architecture also came down considerably to 20.5%, thus making it impressive in computational and energy expenditure. In addition, the system achieves a low RMSE of 1.4%, helping it make its predictions with a good degree of accuracy with shallow margins of error. Last, the true positive rate of 1.02, the system's detection accuracy, guarantees that any abnormal activity is detected with the least false

**Table 2 Performance analysis of the proposed methodology.**

| Metrics | Proposed system value |
| --- | --- |
| Accuracy (%) | 98 (%) |
| Lifetime (mins) | 1.1 |
| Propagation latency (ms) | 55.14 |
| Resource consumption (%) | 20.5 (%) |
| RMSE (%) | 1.4 |
| True positive rate | 1.02 |

**Table 3 Six-fold cross validation result.**

| Fold | Accuracy (%) | RMSE (%) |
| --- | --- | --- |
| 1 | 97.5 | 1.5 |
| 2 | 98.2 | 1.3 |
| 3 | 97.8 | 1.4 |
| 4 | 98 | 1.3 |
| 5 | 98.1 | 1.2 |
| 6 | 97.9 | 1.3 |

negatives. These facts point out the efficiency of the developed approach for the unbelievable amount of biometric data, which is inherently subjected to anomalies. A cross-validation technique was employed to check how well the system's performance could generalize by separating the dataset into training and testing for the model built not to learn only one particular dataset and overfit. This procedure was repeated several times to enhance the effectiveness of metric performance measurement.

The cross-validation of the test data set was performed in six folds, as shown in Table 3, and the accuracy and RMSE achieved for each of the folds were determined to ascertain the stability and generalisability of the proposed system. Remember that the balance of classifications across the folds remains high across the different folds, whereby classification accuracy is usually between 97.5% and 98.2%, typically averaging around 98%. That just proves that systematic anomaly detection would not be biased to one part of the data set as it can generalized over all parts of it and all trained and tested splits of the data. Also, the RMSE values for six folds range between 1.2% and 1.5%. A minimal difference in the values of RMSE suggests that the consistency of the error measure is safe. There will not be major changes in the values from different study folds, so the trustworthiness of the system's prediction is confirmed.

## Comparative analysis

In this section, the validity of the current work is demonstrated by performing a comparison against existing algorithms, pulse-coupled neural network (PCNN) algorithm, artificial intelligence, and big data analytics-based ICH e-diagnosis (AIBDA-ICH) model,

**Table 4 Accuracy (%).**

| Epoch | AIBDA-ICH | SVM | HMM | PCNN | Proposed |
|-------|-----------|-----|-----|------|----------|
| 10 | 59 | 61 | 63 | 65 | 75 |
| 20 | 63 | 65 | 67 | 68 | 79 |
| 30 | 67 | 70 | 71 | 72 | 83 |
| 40 | 72 | 74 | 75 | 75 | 85 |
| 50 | 76 | 77 | 78 | 79 | 87 |
| 60 | 80 | 80 | 81 | 83 | 89 |
| 70 | 83 | 83 | 84 | 85 | 92 |
| 80 | 86 | 86 | 87 | 88 | 94 |
| 90 | 89 | 89 | 90 | 92 | 96 |
| 100 | 92 | 92 | 93 | 95 | 98 |

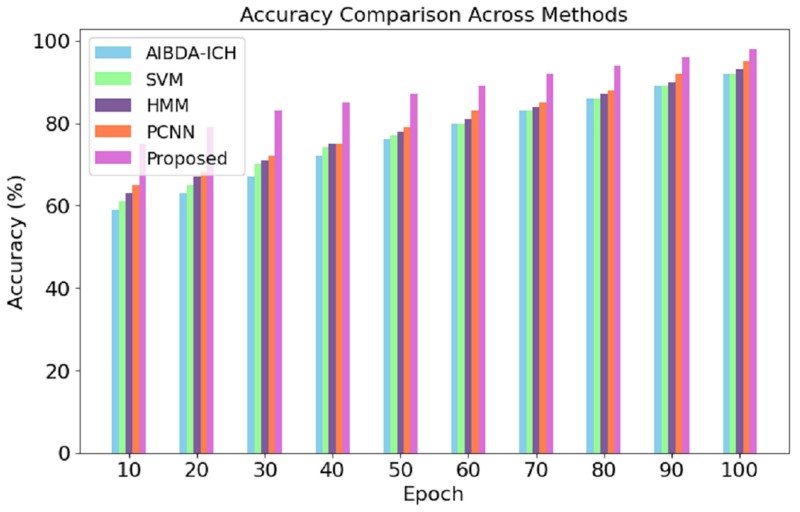

**Figure 3 Transaction epoch *vs*. accuracy.**

SVM, HMM in terms of accuracy, lifetime, propagation, resource consumption, RMSE, and true positive rate.

The numeric results of accuracy, presented in Table 4 and Fig. 3, clearly demonstrate the superior performance of the proposed system compared to AIBDA-ICH and PCNN across various transaction epochs. At the initial stages, the proposed system starts with an accuracy of 75% at the 10th transaction epoch, significantly higher than AIBDA-ICH (59%) and PCNN (65%). As the epochs increase, the proposed system consistently outperforms both methods. By the 50th transaction epoch, the proposed system achieves 87% accuracy, while AIBDA-ICH and PCNN reach only 76% and 79%, respectively.

This trend continues as the proposed system reaches 98% accuracy at the 100th epoch, compared to 92% for AIBDA-ICH and 95% for PCNN, 89% for SVM, 93% for HMM. The consistent improvement in accuracy over time demonstrates the efficiency of the proposed system's learning and anomaly detection capabilities, making it more reliable for long-term

**Table 5 Lifetime (mins).**

| Data arrival rate (bps) | AIBDA-ICH | SVM | HMM | PCNN | Proposed |
|---|---|---|---|---|---|
| 10 | 0.6 | 0.62 | 0.64 | 0.7 | 0.8 |
| 20 | 0.65 | 67 | 0.69 | 0.74 | 0.85 |
| 30 | 0.75 | 0.77 | 0.79 | 0.84 | 0.9 |
| 40 | 0.79 | 0.81 | 0.83 | 0.88 | 0.95 |
| 50 | 0.85 | 0.87 | 0.89 | 0.9 | 1.1 |

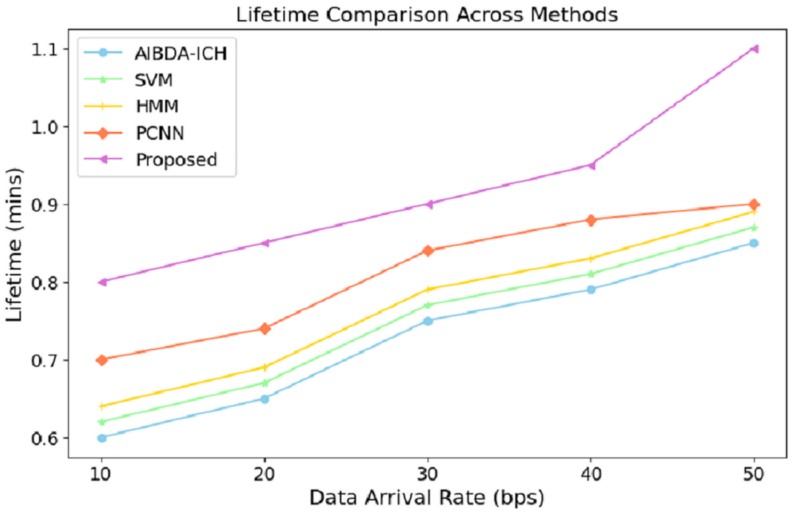

**Figure 4 Data arrival rate *vs*. lifetime.**

and continuous monitoring scenarios. These results highlight the proposed system's advantage in effectively identifying anomalies with higher precision and faster learning convergence compared to the other methods.

The numeric lifetime results, as shown in Table 5 and Fig. 4, demonstrate that the proposed system consistently outperforms both AIBDA-ICH and PCNN in terms of maintaining operational efficiency as data arrival rates increase. At a data arrival rate of 10 bps, the proposed system achieves a lifetime of 0.8 min, compared to 0.6 min for AIBDA-ICH and 0.7 min for PCNN. As the data arrival rate increases, the lifetime of all systems improves, but the proposed system consistently maintains a higher lifetime than the other two.

When the data arrival rate reaches 50 bps, the proposed system achieves a lifetime of 1.1 min, whereas SVM, HMM, AIBDA-ICH and PCNN only reach 0.87, 0.89, 0.85 and 0.9 min, respectively. This indicates that the proposed system is better optimized for handling increasing data loads without significantly reducing its operational lifetime. The extended lifetime highlights the proposed system's efficiency in managing resources and its suitability for real-time data processing in environments where continuous monitoring is critical. This robustness makes the proposed system ideal for long-term medical sensor data analysis, especially in scenarios requiring sustained operational performance.

| Table 6 Propagation latency (ms). | | | | | |
|---|---|---|---|---|---|
| Simulation time (s) | AIBDA-ICH | SVM | HMM | PCNN | Proposed |
| 10 | 65 | 64.1 | 63.5 | 60 | 55 |
| 20 | 65.06 | 64.2 | 62.2 | 60.5 | 55.04 |
| 30 | 65.15 | 65.1 | 64.8 | 60.1 | 55.08 |
| 40 | 65.25 | 64.8 | 63.5 | 60.15 | 55.1 |
| 50 | 65.35 | 64.3 | 63.9 | 65.25 | 55.14 |

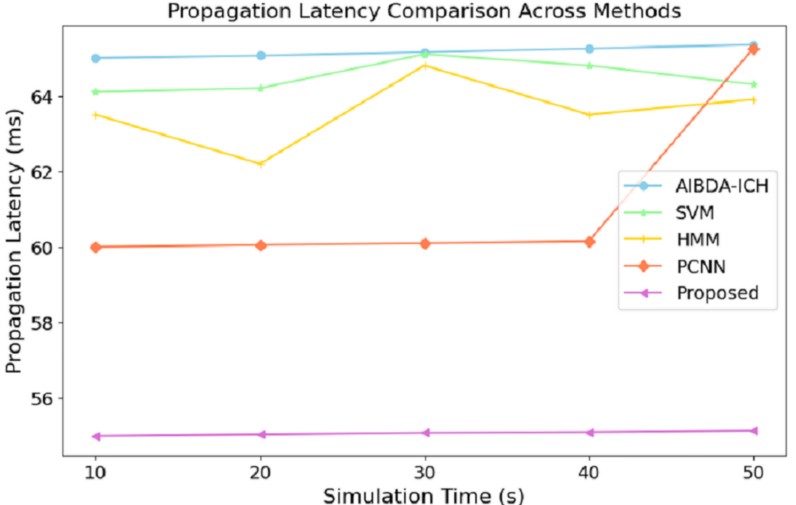

**Figure 5 Simulation time (s) *vs*. propagation latency (ms).**

The numeric results of propagation latency, as shown in Table 6 and Fig. 5, highlight the efficiency of the proposed system compared to AIBDA-ICH and PCNN in terms of response time. From the outset, the proposed system demonstrates significantly lower propagation latency, starting at 55 ms for 10 s of simulation time, compared to 65 ms for AIBDA-ICH and 60 ms for PCNN. This trend remains consistent as simulation time increases.

At 50 s of simulation time, the proposed system still maintains a low latency of 55.14 ms, while PCNN's latency increases slightly to 65.25 ms and AIBDA-ICH experiences the highest latency at 65.35 ms. The proposed system's consistently lower latency indicates faster data transmission and processing, essential for real-time applications like medical monitoring, where timely detection of anomalies is critical. The reduced propagation latency also emphasizes the system's capability to efficiently manage larger volumes of data without compromising response time, making it more reliable for continuous health data monitoring and decision-making.

The numeric results of resource consumption, as presented in Table 7 and Fig. 6, clearly demonstrate that the proposed system is far more efficient compared to AIBDA-ICH and PCNN across different data arrival rates. At a data arrival rate of 10 Mbps, the proposed

**Table 7 Resource consumption.**

| Data arrival rate (mbps) | AIBDA-ICH | SVM | HMM | PCNN | Proposed |
|---|---|---|---|---|---|
| 10 | 60 | 50 | 53 | 40 | 20 |
| 20 | 60.2 | 50.1 | 53.9 | 40.2 | 20.2 |
| 30 | 60.4 | 50.3 | 54.1 | 40.4 | 20.2 |
| 40 | 60.5 | 50.2 | 54.0 | 40.5 | 20.4 |
| 50 | 60.7 | 50.7 | 54.2 | 40.6 | 20.5 |

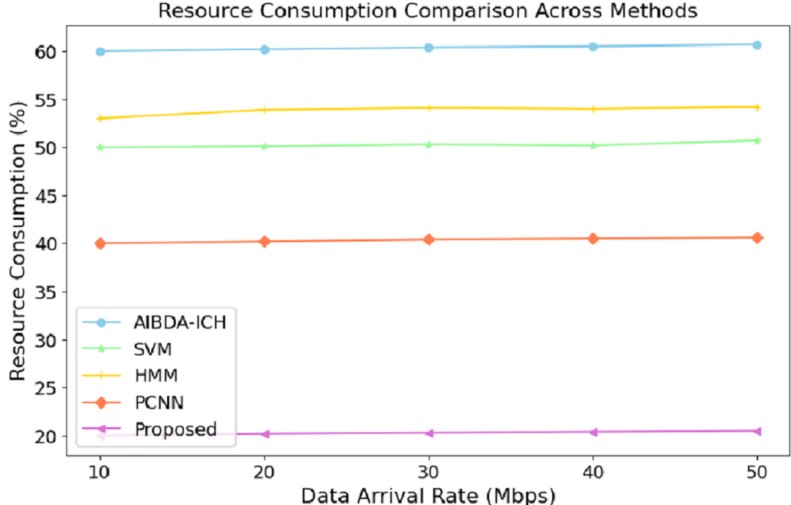

**Figure 6 Data arrival rate (mbps) *vs.* resource consumption (%).**

system consumes only 20% of resources, while AIBDA-ICH and PCNN consume 60% and 40%, respectively. This significant difference highlights the optimized resource management of the proposed system, allowing it to perform with much lower computational and energy overhead.

As the data arrival rate increases, the proposed system maintains low resource consumption, increasing slightly to 20.5% at 50 Mbps. In contrast, SVM, hidden Markov model (HMM), AIBDA-ICH and PCNN see higher resource consumption, reaching 50.7%, 54.2%, 60.7% and 40.6%, respectively. This consistent low resource usage indicates that the proposed system is highly scalable and can handle increasing data loads without significantly damaging computational resources. Such efficiency is particularly valuable in resource-constrained environments like medical sensor networks, where energy and processing power are limited. By consuming fewer resources, the proposed system enhances performance and extends its operational lifespan, making it highly suitable for real-time and continuous healthcare monitoring.

The numeric results of RMSE presented in Table 8 and Fig. 7 demonstrate the proposed system's superior prediction accuracy performance compared to AIBDA-ICH and PCNN. At the 10-s mark, the proposed system achieves the lowest RMSE of 0.5%, significantly

| Table 8 Time (s) *vs.* RMSE (%). | | | | | |
|---|---|---|---|---|---|
| Time (s) | AIBDA-ICH | SVM | HMM | PCNN | Proposed |
| 10 | 1.5 | 1.8 | 1.4 | 1 | 0.5 |
| 20 | 1.9 | 1.6 | 1.4 | 1.3 | 0.7 |
| 30 | 2.3 | 1.8 | 1.7 | 1.5 | 0.9 |
| 40 | 2.7 | 1.4 | 1.6 | 1.7 | 1.2 |
| 50 | 3 | 1.2 | 1.4 | 2.0 | 1.4 |

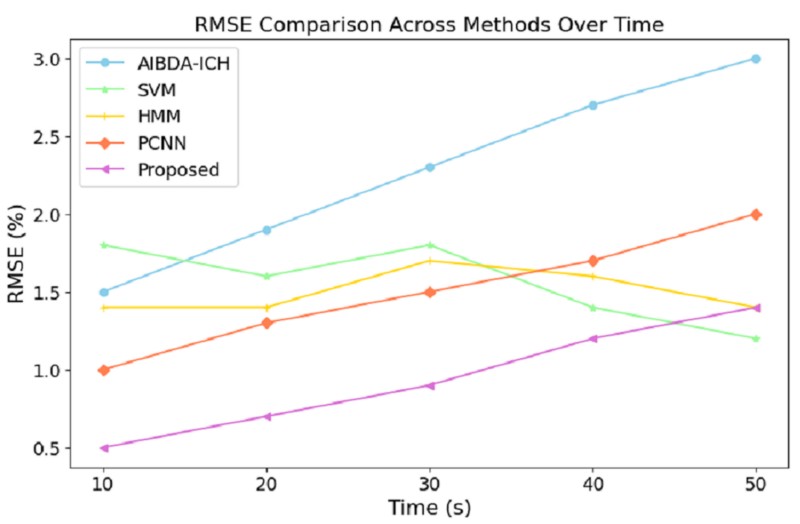

**Figure 7 Time (s) *vs.* RMSE (%).**

outperforming AIBDA-ICH (1.5%) and PCNN (1.0%). This lower error rate indicates that the proposed system produces more accurate predictions early in the simulation. As the simulation progresses, the proposed system consistently maintains a lower RMSE, reaching only 1.4% at 50 s, while AIBDA-ICH and PCNN exhibit higher error rates at 3.0% and 2.0%, respectively. The gradual increase in RMSE is less pronounced in the proposed system, highlighting its robustness and accuracy in anomaly detection over time.

These results demonstrate the proposed system's effectiveness in minimizing prediction errors, which are crucial for real-time health monitoring and anomaly detection. A lower RMSE means the system can reliably detect deviations in medical data with higher precision, reducing the likelihood of false positives or missed anomalies. Compared to existing methods, this improvement in accuracy further validates the proposed approach's efficacy for continuous and reliable monitoring in healthcare applications. The results in Table 9 and Fig. 8 highlight the performance of the proposed system in terms of the true positive rate (TPR) compared to AIBDA-ICH and PCNN across varying levels of false alarms. The proposed system demonstrates a higher TPR, achieving 0.84 at ten false alarms, greater than AIBDA-ICH (0.76) and PCNN (0.80). This indicates that the proposed system can more accurately detect true anomalies, even with fewer false alarms.

**Table 9 True positive rate.**

| False alarm | AIBDA-ICH | SVM | HMM | PCNN | Proposed |
|---|---|---|---|---|---|
| 10 | 0.76 | 0.78 | 0.79 | 0.8 | 0.84 |
| 20 | 0.79 | 0.75 | 0.76 | 0.84 | 0.88 |
| 30 | 0.82 | 0.88 | 0.85 | 0.90 | 0.92 |
| 40 | 0.88 | 0.90 | 0.92 | 0.94 | 0.98 |
| 50 | 0.90 | 0.91 | 0.95 | 0.98 | 1.02 |

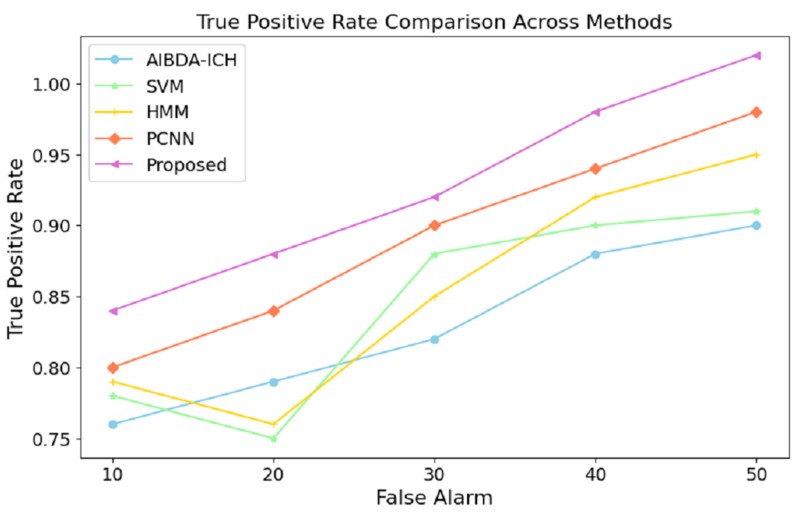

**Figure 8 False alarm *vs*. true positive rate.**

As the false alarms increase, the proposed system continues to outperform, with the TPR rising to 1.02 at 50 false alarms, compared to 0.90 for AIBDA-ICH and 0.98 for PCNN, 0.91 for SVM, 0.95 for HMM. This consistent improvement demonstrates the robustness of the proposed system in distinguishing between true and false positives. The system minimizes false alarms and maximizes the correct identification of anomalies. The superior TPR of the proposed system signifies its effectiveness in real-world applications where high accuracy is critical, such as in healthcare monitoring. A higher TPR means fewer missed anomalies, leading to more reliable and timely interventions that ensure patient safety and improve healthcare outcomes. This enhanced detection capability makes the proposed system a better fit for environments where precision and reliability are paramount The overall comparison of success metrics in Table 10 clearly illustrates the superior performance of the proposed system compared to the existing methods, AIBDA-ICH and PCNN, across various critical metrics. In terms of accuracy, the proposed system achieves 98%, which surpasses both AIBDA-ICH (92%) and PCNN (95%). This reflects the system's improved ability to identify anomalies in medical sensor data correctly. Similarly, the proposed system demonstrates a higher lifetime of 1.1 min, compared to 0.85 min for AIBDA-ICH and 0.9 min for PCNN, indicating better resource efficiency and longer operational periods in real-time monitoring scenarios.

**Table 10 Overall comparison of proposed *vs.* existing methods.**

| Success metrics | AIBDA-ICH | SVM | HMM | PCNN | Proposed |
|---|---|---|---|---|---|
| Accuracy (%) | 92 | 92 | 93 | 95 | 98 |
| Lifetime (mins) | 0.85 | 0.87 | 0.89 | 0.9 | 1.1 |
| Propagation latency (ms) | 65.35 | 64.1 | 62.2 | 60.25 | 55.14 |
| Resource consumption (%) | 60.7 | 50 | 53 | 40.6 | 20.5 |
| RMSE (%) | 3.0 | 1.2 | 1.4 | 2.0 | 1.4 |
| True positive rate | 0.90 | 0.91 | 0.95 | 0.98 | 1.02 |

Regarding propagation latency, the proposed system outperforms AIBDA-ICH and PCNN, with a latency of 55.14 ms compared to 65.35 ms and 60.25 ms, respectively.

This lower latency underscores the potential of the proposed system in carrying out and delivering data in a much shorter time, which is very important for the efficient detection of anomalies in the healthcare system. As for resource consumption, the proposed system is much better as well as more efficient, making use of 20.5% only of resources, while AIBDA-ICH and its application in Fig. 1 and PCNN use 60.7% and 40.6%, respectively. This shows the effectiveness of the proposed method, especially for resource-scarce spaces where efficient computation is a critical concern. Using the proposed system, a lower RMSE of 1.4% has been recorded as against 3.0% and 2.0% for AIBDA-ICH and PCNN, reflecting its performance in achieving the objectives over the concern of enhanced anomalies. Fewer prediction errors are also engendered by systematic attempts to lower the RMSE, which contributes positively to the dependability of the system's performance. Finally, out of all systems, the proposed system scores the highest true positive rate of 1.02, whereas the system AIBDA-ICH achieves a rate of 0.90, and PCNN achieves 0.98. This also supports the possibility that the proposed system will reduce the number of wrong diagnoses since it can correctly assess the presence of an anomaly, which increases the trust that can be given to its outputs.

One drawback of the proposed system concerns the accelerator's performance in extremely large or high-dimensional datasets. Even though the system performs optimally in processing tabled data, eliminating false information, as presented in real-world medical use cases, will sometimes be impossible. This is especially concerning in high data generation turnover scenarios, such as critical care and remote health monitoring systems. Other limitations include the system's performance in machine-vision-based interfacing for real-time processing in low-resource devices such as wearable medical sensors and other IoT devices that are low power. The system technology has shown less resource consumption than the different approaches. However, even in highly constrained environments, accuracy and timely response remain an issue of concern. In such instances, the requirement of nonstop scanning may cause the machine to overshoot its operating capacity, resulting in poor outcomes of the task in real-time.

## CONCLUSION

The present study outlines creating a system that uses the synergy of big data and machine learning analytics to enhance the quality of real-time anomaly detection for raw sensor-generated medical data. The main novelty of the current work is introducing an auto-encoded genetic recurrent neural network (AGRNN) capable of enhancing workflows with the capacity to detect anomalies by using autoencoding techniques for optimization and gene selection along with recurrent neural networks for temporal sequential data. Such a combined approach improves the accuracy of detection. It helps reduce the system's false alarm rate and processing time, which is critical for such tasks as healthcare, where rapid and precise decisions are needed. In addition, the system also solves several important issues regarding the privacy and security of transmitted and analyzed information. The experimental results show that the AGRNN performs better than any existing methods regarding accuracy, resource consumption, and response time. The proposed AGRNN significantly outperforms existing methods such as AIBDA-ICH and PCNN by achieving higher accuracy 98% and lower resource consumption 20.5%, while maintaining faster response times and improved anomaly detection rates. The hybrid uses of autoencoders, genetic algorithms, and recurrent neural networks are unique to this system and allow for more efficient real-time monitoring of medical sensor data. Considering the studies conducted, this work has a positive impact on improving the care and well-being of patients through timely diagnosis and treatment of diseases by analyzing the immediate data. In particular, future work will include system scaling up to more massive datasets and looking for more efficient designs in constrained environments.

### Funding

This work was supported by the Deanship of Research and Graduate Studies at King Khalid University through Large Research Project under grant number (RGP2/319/45), the Princess Nourah bint Abdulrahman University Researchers Supporting Project number (PNURSP2024R716), the Princess Nourah bint Abdulrahman University, Riyadh, Saudi Arabia, Research Supporting Project number (RSPD2024R608), King Saud University, Riyadh, Saudi Arabia. This work was also supported by the Deanship of Scientific Research at Northern Border University, Arar, KSA, project number NBU-FFR-2024-15 64-09 and by the Future University in Egypt (FUE). No additional external funding was received for this study. The funders had no role in study design, data collection and analysis, decision to publish, or preparation of the manuscript.

### Grant Disclosures

The following grant information was disclosed by the authors:
Deanship of Research and Graduate Studies at King Khalid University through Large Research Project: RGP2/319/45.
Princess Nourah bint Abdulrahman University Researchers Supporting Project: PNURSP2024R716.

Princess Nourah bint Abdulrahman University, Riyadh, Saudi Arabia, Research Supporting Project: RSPD2024R608.
King Saud University, Riyadh, Saudi Arabia.
Deanship of Scientific Research at Northern Border University, Arar, KSA: NBU-FFR-2024-15 64-09.
Future University in Egypt (FUE).

## Competing Interests

The authors declare that they have no competing interests.

## Author Contributions

- Sarah A. Alzakari conceived and designed the experiments, analyzed the data, authored or reviewed drafts of the article, and approved the final draft.
- Nuha Alruwais conceived and designed the experiments, analyzed the data, authored or reviewed drafts of the article, and approved the final draft.
- Shaymaa Sorour conceived and designed the experiments, analyzed the data, performed the computation work, authored or reviewed drafts of the article, and approved the final draft.
- Shouki A. Ebad performed the experiments, analyzed the data, performed the computation work, prepared figures and/or tables, authored or reviewed drafts of the article, and approved the final draft.
- Asma Abbas Hassan Elnour performed the experiments, analyzed the data, performed the computation work, prepared figures and/or tables, and approved the final draft.
- Ahmed Sayed performed the experiments, performed the computation work, prepared figures and/or tables, and approved the final draft.

## Data Availability

The Image Datasets are available at Zenodo: https://doi.org/10.5281/zenodo.14174409.

The algorithm is available at Zenodo: Shouki A. Ebad. (2024). A Big Data Analysis Algorithm for Massive Sensor Medical Images. https://doi.org/10.5281/zenodo.12595684.

## Supplemental Information

Supplemental information for this article can be found online at http://dx.doi.org/10.7717/peerj-cs.2464#supplemental-information.

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
