# Peer review of "A big data analysis algorithm for massive sensor medical images"

_PeerJ Computer Science, doi:10.7717/peerj-cs.2464_

## Round 0.1 · original submission · Major Revisions

Thank you for the opportunity to consider your work. Please address the review comments and resubmit.

·

Basic reporting

The manuscript is difficult to follow.
It is not clear what is innovative part.
How and where big data analysis was used?
The manuscript is full of verz well known issues but it does not look connected.
Finally I did not catch which algorithm was used as proposed.
Comparison with AIBDA-ICH, PCNN is not clear.
There are lot of information but no clear way of the main contribution of this manuscript.
What are limitations?
How data were collected?

Experimental design

Not clear.
DIfficult to understand.

Validity of the findings

I did not see validation of the findings.

Additional comments

This manuscript needs very much corrections.

Cite this review as

Reviewer 2 ·

Basic reporting

The advantages of the proposed scheme over existing schemes are unclear. Many details in the article cannot be seen whether the original design of the article or the reference to existing methods. For example, the introduction of the neural network used in the paper is not clear.

Experimental design

The experimental part of the manuscript is inadequate. The author only compares two methods, and PCNN is an old one. The authors need to compare with more recent methods.

Validity of the findings

no comment

Additional comments

The images are blurry and the format of the formulas is difficult to read.

Cite this review as

Reviewer 3 ·

Basic reporting

The authors present an article titled as A Big Data Analysis Algorithm for Massive Sensor Medical Images.
1. I am not convinced about the novelty of the manuscript. The novelty of the paper needs to be justified and clearly defined. It includes a clear difference between the available literature and previous works. The authors are asked to provide the limitations of the previous correlated works and then link those limitations to the current ideas and contributions of the current work.
2. The authors should clearly mention the challenges in the abstract. Mention a few recent techniques and highlight what are the challenges faced by the systems and then present the proposed objective. Objectives and challenges are quite different from each other.
3. Expand the critical results in the conclusion. Focus on the main developments in the finale. Also, write the main contributions in the conclusion.
4.Measure the performance of the model with various parameters and discuss the observations in detail.

Experimental design

4.Measure the performance of the model with various parameters and discuss the observations in detail.

Validity of the findings

3. Expand the critical results in the conclusion. Focus on the main developments in the finale. Also, write the main contributions in the conclusion.

Cite this review as

---

## Round 0.2 · accepted · Accept

Dear authors, we are pleased to verify that you meet the reviewer's valuable feedback to improve your research.

Thank you for considering PeerJ Computer Science and submitting your work.

Reviewer 2 ·

Basic reporting

no comment

Experimental design

no comment

Validity of the findings

no comment

Cite this review as

Reviewer 3 ·

Basic reporting

All comments are addressed

Experimental design

All comments are addressed

Validity of the findings

All comments are addressed

Additional comments

All comments are addressed

Cite this review as